# Process Parameters and Geometry Effects on Piezoresistivity in Additively Manufactured Polymer Sensors

**DOI:** 10.3390/polym15092159

**Published:** 2023-04-30

**Authors:** Marijn Goutier, Karl Hilbig, Thomas Vietor, Markus Böl

**Affiliations:** 1Institute for Engineering Design, Technische Universität Braunschweig, 38108 Brunswick, Germany; k.hilbig@tu-braunschweig.de (K.H.); t.vietor@tu-braunschweig.de (T.V.); 2Institute of Mechanics and Adaptronics, Technische Universität Braunschweig, 38106 Brunswick, Germany; m.boel@tu-braunschweig.de

**Keywords:** conductive polymer composite, resistive sensor, piezoresistive sensor, process parameters, design for additive manufacturing, conductive filament, additive manufacturing, 3D printing, material extrusion, fused deposition modeling

## Abstract

The current work experimentally determined how the initial resistance and gauge factor in additively manufactured piezoresistive sensors are affected by the material, design, and process parameters. This was achieved through the tensile testing of sensors manufactured with different infill angles, layer heights, and sensor thicknesses using two conductive polymer composites. Linear regression models were then used to analyze which of the input parameters had significant effects on the sensor properties and which interaction effects existed. The findings demonstrated that the initial resistance in both materials was strongly dependent on the sensor geometry, decreasing as the cross-sectional area was increased. The resistance was also significantly influenced by the layer height and the infill angle, with the best variants achieving a resistance that was, on average, 22.3% to 66.5% lower than less-favorable combinations, depending on the material. The gauge factor was most significantly affected by the infill angle and, depending on the material, by the layer height. Of particular interest was the finding that increasing in the infill angle resulted in an increase in the sensitivity that outweighed the associated increase in the initial resistance, thereby improving the gauge factor by 30.7% to 114.6%, depending on the material.

## 1. Introduction

Additive manufacturing (AM) is a family of manufacturing processes in which a part is produced by adding material, typically in a layer-by-layer fashion. These types of manufacturing processes have several advantages over traditional manufacturing methods, such as the ability to combine multiple materials within a single part and to produce highly complex geometry without a significant cost impact [1]. The most commonly used AM processes are vat photopolymerization, material extrusion (MEX), and powder bed fusion for both metals and polymers [2,3]. Of these processes, MEX is the most capable of producing multi-material components, as multiple extruders can be easily implemented. In comparison, the other two commonly used AM processes require liquid polymer vats and powder beds, respectively, where multi-material parts are more difficult to achieve. MEX also has a wide range of compatible commercially available thermoplastics and has attracted attention for the possibility of using recycled filaments [4,5,6].

The multi-material capabilities of MEX allow for the local adjustment of mechanical properties and have also proven useful in enhancing part functionality, for example by enabling the production of integrated electronics and sensors [7]. The manufacturing of electronics using MEX can be broadly divided into direct and indirect production methods. Examples of indirect methods include interrupting the AM process to embed pre-fabricated conductors or electronic components [8] or achieving conductivity through post-processing steps, such as infusing channels with a conductive liquid [9] or via selective metal plating [10]. The direct methods for achieving conductivity in AM parts use materials that are inherently conductive, without the need for prefabricated components or post-processing steps. When using MEX, the appropriate materials for the direct AM of electronics are conductive polymer composite (CPC) materials. These consist of a conductive filler in a non-conductive polymer matrix. Various fillers have been reported to be compatible with MEX, such as carbon nano tubes (CNTs) [11,12], graphene [13], carbon black (CB) [14], and copper nano-wires [15]. Such fillers can be combined with one of the numerous polymers compatible with the MEX processes, although the creation of polymer composites suitable for AM does necessitate the consideration of several requirements. Conductivity, melt viscosity, and elastic modulus typically increase with a higher filler content, but the latter two can make filament preparation and its subsequent use challenging [16]. Certain fillers also exhibit a tendency to agglomerate within the matrix material, negatively affecting both the mechanical and electrical properties [16,17]. In addition, there is a need for both the matrix and filler materials to be compatible with the required processing temperatures, with copper nano-wires in particular suffering from oxidation and reduced conductivity when exposed to high temperatures [15]. Some examples of successfully implemented polymers include polylactic acid (PLA) and thermoplastic polyurethane (TPU), the latter of which is particularly capable of withstanding high deformation in soft and flexible electronics [11,12,18].

Conductive polymer composites can be used to fabricate a wide variety of electronics and sensors [19,20]. A review by Ni et al. [21] describes the use of MEX for flow rate, strain, tactile, and temperature sensors. While AM sensors can operate based on a variety of underlying phenomena, sensors for measuring mechanical loads are most commonly based on either the capacitive or the piezoresistive effect [22]. In the case of piezoresistive sensors, the conductive filler particles form a percolation network that exhibits a change in electrical resistance when subjected to strain [23].

Table 1 provides an overview of a selection from the existing literature on sensors for mechanical loads produced using MEX. These are separated according to the load cases investigated and according to the main focus of the research.

One of the earliest attempts to develop a conductive polymer composite specifically for AM is presented by Leigh et al. [7]. This work focuses on the formulation requirements of a polycaprolactone (PCL) with CB composite and presents a number of sensor concepts. Examples include a piezoresistive sensor subjected to bending and capacitive buttons subjected to compression.

There are a number of further examples in the current literature that focus on the influence of the material composition on the conductivity or sensor characteristics. The work of Christ et al. [11] presents TPU filaments with different CNT concentrations and tests how their conductivity and piezoresistive behavior are affected. Comparisons are made between the behavior of the bulk material and after additive manufacturing, but the influence of the AM process parameters is not investigated. Kim et al. [30] present different compositions of TPU with CNT fillers, which are used to fabricate piezoresistive sensors in a single beam and in a multiaxial 3D layout. Similarly, Hohimer et al. [31] investigate TPU filaments with different CNT concentrations and use them to fabricate capacitive sensors and flexible piezoresistive self-sensing actuators. Georgopoulou et al. [25] present a soft robotic gripper and measure the piezoresistive behavior under a tensile load for two variants of TPU with CB, each with a different hardness. The properties of hybrid fillers are investigated by Xiang et al. using blends of CNTs and graphene [13] and CNTs and silver nanoparticles [24] within a TPU matrix.

There is also work that focuses on the influence of process parameters on conductivity and sensor properties. Dijkshoorn et al. [14] present a sensor using PLA with a CB filler and investigate the effects of different build orientations and infill patterns on the resistivity. While the effects of these parameters on resistivity are reported, the work does not provide insight into the influences on the piezoresistive behavior. Arh et al. [26] perform dynamic measurements on PLA sensors with CB fillers and compare the effect of different build orientations on the piezoresistive coefficient. A piezoresistive sensor produced using CNTs in an acrylonitrile butadiene styrene (ABS) matrix is presented by Dul et al. [27]. Samples with two different infill patterns are produced, and the sensors are characterized under different loading conditions. Munasinghe et al. [29] develop a strain sensor for load and wear sensing in industrial equipment, which is made of a PLA matrix with CB filler. The sensors are printed with 45° and 90° infill angles, but the effects of this variation are only described in terms of stiffness and strength, not the piezoresistive properties. Stano et al. [32] present a design of experiments approach, in which the effect of part orientation, line width, and layer height are varied to determine the combination that results in the lowest resistance in an unloaded state. The lowest-resistance variants are then used to construct a load cell [33]. In particular, it is assumed that the combination of parameters that results in the lowest resistance is the most suitable, although no experiments are performed to determine if the sensor response under load is affected by the choice of process parameters.

The geometric design freedom that exists when using AM has inspired a number of works focusing on the influence of a sensor’s geometry on its properties. Christ et al. [12] investigate the piezoresistive behavior as realized by using different meandering sensor patterns. Mousavi et al. [28] create sensors with an anisotropic response by tailoring the geometry, infill ratio, and bed temperature. This anisotropy allows the detection of the load direction as well as an increased piezoresistive response. Watschke et al. [22] present a work that focuses on the influences of different geometric variants on a piezoresistive sensor design that operates via a resistive-path-shortening mechanism under compressive loading. Maurizi et al. [34] investigate the dynamic properties of PLA sensors with CB fillers. Variants of different lengths are characterized while keeping the material and process parameters constant. The geometry is found to have an influence on the measurement range, gauge factor, and signal linearity, while an attempt to compensate for electromagnetic noise by including an additional sensor on the neutral axis is reported as unsuccessful. Schouten et al. [35] demonstrate the use of the sensing element arrangement as a means of signal linearization. A flexible TPU with a CB filler sensor is designed so that two sensing elements are oppositely loaded under a bending load. The differential measurement of these two signals provides a sensor output with improved linearity compared to a single sensing element.

While the feasibility of the additive manufacturing of sensors using MEX has been demonstrated, there remains an incomplete understanding of the effect of the sensor geometry, AM process parameters, material selection, and their interactions on the sensor characteristics. The current work aims to improve the understanding of the various influences on AM piezoresistive sensors by investigating how the sensor resistance and sensitivity are affected as the AM process settings and sensor geometry are varied. This was be accomplished through the tensile testing of sensors with varying thicknesses, infill angles, layer heights, and materials. Some factors, such as the effect of the layer height on the sensitivity, were completely unexplored, while other factors were only studied in isolation, missing potential interaction effects. Understanding the drivers of sensor behavior is important to enable the design of sensors with desired properties, bringing them closer to practical applications. Identifying the significant parameters is also valuable for future research, as reporting them in the literature is essential for describing reproducible experiments.

## 2. Materials and Methods

### 2.1. Sensor Design and AM Setup

Piezoresistive sensors consisting of a conductive sensing element and a non-conductive carrier material were manufactured using MEX. The non-conductive part of each sensor was made from NinjaTek^®^ NinjaFlex filament [36], a flexible thermoplastic polyurethane (TPU). This non-conductive carrier allowed the overall geometry to remain constant while the height of the conductive sensing element could be varied. It also ensured that the conductive part of the sensor was not printed in direct contact with the print bed, where the properties often differ from other locations of an AM part.

The conductive elements were made of either a conductive thermoplastic polyurethane (cTPU) or a conductive polylactic acid (cPLA), as shown in Table 2. These materials were selected because they are among the most commonly used commercially available conductive filaments in the current literature. The cTPU material is particularly interesting for applications in flexible electronics due to its maximum strain of 355% according to the manufacturer’s specification [37]. The cPLA material allowed a direct comparison to determine if the findings were consistent across the different materials.

The overall sensor geometry was chosen according to test specimen variant A22, as defined in ISO 20753 [41]. A render of such a sensor is shown in Figure 1. The height of the conductor could be varied, as shown in Table 3, while its width was 5.0 mm. The sensors were manufactured flat as shown, i.e., with the conductor facing upwards.

Table 3 lists the four input factors in the experiment: The conductive polymer composite material, the conductor height, and the process parameters layer height and infill angle. Of these factors, the conductor height represented the total height of the sensing element, while the layer height was the thickness of a single layer in the z-axis direction in the additive manufacturing. The infill pattern was, in all cases, a rectilinear pattern rotated to the angle given in Table 3, where 0° was the longitudinal orientation of the sensor. The non-conductive carrier material was produced in all samples with a ±45° infill and a layer height of 0.2 mm. The input factors, as listed in Table 3, resulted in 36 unique combinations. Each of these combinations was produced in triplicate to provide data on process variation and repeatability.

To eliminate the influences of contact resistance, a four-terminal resistance measurement was used. Each sensor therefore had four contact points, as shown in Figure 1. These electrical connection points were treated with colloidal silver paste (type 12640; Electron Microscopy Sciences, Hatfield, PA, USA), which ensured a reliable contact [42]. The outer two contact points were contacted using pogo pins. The two inner contact points were located in between the clamps when in the tensile test setup, as shown in Figure 2. This position ensured that the clamping forces did not affect the measured resistance. These inner points were contacted by attaching copper wires to the colloidal silver paste using a conductive silver epoxy adhesive (8331-14G; MG Chemicals, Burlington, ON, Canada). Flexible wires with a cross-sectional area of 0.14 mm^2^ were chosen to minimize the effect of their stiffness on the measurements. The length of the conductive element between the inner contacts, i.e., the active length where the resistance was measured, was 30 mm.

In accordance with recommendations in the literature [43], the sensors were produced on a material extrusion machine equipped with a tool changer that could switch between multiple tool heads. The machine used was a ToolChanger and Motion System equipped with Hemera direct extruders (E3D-Online Ltd., London, UK). To prevent deviations due to abrasive tool wear when processing the composite materials [44], hardened nozzles with a 0.4 mm diameter were used for all materials (Nozzle X, E3D-Online Ltd., London, UK). These nozzles had a lower thermal conductivity than a typical brass nozzle, requiring an increased temperature setting. The print bed temperature was set at 40 °C, and no cooling fan was used on the parts. The extrusion temperature, speed and flow rate (or extrusion multiplier) were calibrated individually for each material using test specimens, selecting settings that resulted in dense parts with a low electrical resistance. Table 4 summarizes the process settings as used for each material. To accurately measure the effect of the infill patterns, no shells (i.e., shape outlines) were used for the sensing elements. The AM machine instructions were generated using the Simplify3D slicer software (v4.1.2, Simplify3D Inc., Cincinnati, OH, USA).

### 2.2. Measurement Setup

Tensile tests, based on ISO 527-1 [45], were performed on a Zwick Z0.5 axial materials testing machine (Zwick GmbH & Co., Ulm, Germany) with the measurement setup shown in Figure 2. Forces were measured using a 100N Xforce HP load cell (Zwick GmbH & Co., Ulm, Germany), with an accuracy class of 0.5 according to ISO 7500-1. Strain was measured using a VideoXtens 1-120 non-contact video extensometer (Zwick GmbH & Co., Ulm, Germany), with an accuracy class of 1 according to ISO 9513. Measurements were performed in a climate-controlled environment, and the temperature was monitored throughout the experiment to ensure that there were no significant thermoresistive effects. The electrical resistance of the sensors was measured using a Keithley 2750 multimeter/switch system (Tektronix Inc., Beaverton, OR, USA) in a four-point measurement setup.

Initial measurements were made to determine the yield point for each material. This was found to be lowest for cPLA, with results ranging from 1.5% to 1.65% strain. To avoid testing in the plastic deformation range [23], a maximum strain of 0.75% was selected for all experiments. The strain rate was set at 2 mm/min.

### 2.3. Data Analysis Method

The data collected during the experiments was analyzed using Python scripts to determine the following sensor properties:The initial resistance *R_0_* for each sensor in the unloaded state.The sensor sensitivity, as expressed by the dimensionless gauge factor (GF), according to the following equation [46]:
(1)$$GF=\frac{\Delta R/R_{0}}{\varepsilon },$$
where ΔR denotes the change in the sensor resistance, *R*_0_ is the initial resistance, and ε defines the strain. The GF was calculated using resistance values obtained at 0.75% strain.

The values for each sensor are reported in Appendix A in the Appendix A. Once the above sensor characteristics were determined, Minitab (v21.3.1, Minitab LLC., State College, PA, USA) was used to analyze the results. First, graphs of the mean and standard deviation of the sensor characteristics were generated. These graphs provided a visual representation of the relationship between the sensor properties and the experimental variables (material, infill angle, layer height, and conductor height).

Linear regression models were then used to provide a more in-depth analysis of the influencing factors, potential higher-order interactions, and to verify which factors were statistically significant. Separate models were used for each material so that the behavior could be compared between the two. The infill angle was considered as a categorical factor. Data were standardized by subtracting the mean and dividing by the standard deviation. Insignificant terms (*p* > 0.05) were removed from the model through a process of backward elimination while maintaining the model hierarchy. The results of each linear regression model were presented first as a Pareto plot, showing which terms contributed most to the variability of the response in an absolute sense, and second as factorial plots showing the directionality of the main effects.

## 3. Results

### 3.1. Initial Resistance

#### 3.1.1. cPLA

Figure 3 shows a graph of the average initial resistance in the cPLA sensors as a function of the experimental variables. There was a visible difference between the sensors with different conductor heights, while a smaller difference existed as a function of the layer height.

The linear regression model for the initial resistance achieved an R-squared value of 99.30%. Figure 4 shows the Pareto plot for the significant model terms. The terms with the largest impact were consistent with the observations from Figure 3; the conductor height had the largest impact, although its behavior was best captured by including a quadratic term. The effect of the layer height was also found to be statistically significant. Finally, the infill was also found to be significant and interacted with each of the previous factors.

While the Pareto plot in Figure 4 provides absolute values, Figure 5 provides factorial plots for the main effects on the initial resistance, visualizing the directionality of the effect.

#### 3.1.2. cTPU

Figure 6 shows a graphical representation of the average initial resistance in the cTPU sensors in relation to the experimental variables. Compared to the cPLA sensors, the cTPU sensors had a higher initial resistance. Similar to the cPLA, an increased conductor height resulted in a lower initial resistance. However, there appeared to be a greater influence of both the layer height and the infill angle than in the cPLA.

The regression model for the initial resistance in the cTPU had an R-squared value of 98.74%. Figure 7 shows the Pareto plot for the statistically significant terms. The conductor height again had the largest impact, but when compared to the cPLA, the model confirmed that the influences of the layer height and infill parameters were more pronounced. There was also a significant interaction effect between the layer height and the conductor height, which was not present in the cPLA, as well as a higher-order interaction of this effect with the infill angle.

Figure 8 shows factorial plots for the main effects on the initial resistance, visualizing the directionality of the effects.

### 3.2. Gauge Factor

#### 3.2.1. cPLA

Figure 9 provides a graphical representation of the mean gauge factor in the cPLA sensors, separated according to the experimental parameters. There was an increase related to the infill angle, while neither the layer height nor the conductor height seemed to have a clear trend.

The linear regression model for the gauge factor in the cPLA had an R-squared value of 77.87%. In this model, the infill was found to be the only significant contributor to the variance in the gauge factor. The layer height, conductor height, higher-order effects, and interaction effects were all found to have no statistically significant effect. The Pareto plot was omitted in this case because there was only one significant effect.

The factorial plot shown in Figure 10 illustrates the effect of the infill angle. The gauge factor increased with the infill angle, with a 90° infill angle having a mean gauge factor of 24.32, an increase of 30.7% over the 18.61 gauge factor that occurred with a 0° infill.

#### 3.2.2. cTPU

Figure 11 shows a graph of the mean gauge factor for the cTPU sensors separated by experimental variables. Similar to the cPLA, there was an increase in the GF associated with the infill angle. There also appeared to be a more pronounced effect of the layer height, although the direction was not consistent for all the variants.

The regression model for the gauge factor in the cTPU had an R-squared value of 81.94%. The Pareto plot in Figure 12 confirms that the infill angle was the most significant factor affecting the gauge factor. This was followed by a three-way interaction between each of the main model terms and the layer height and its interaction effects. The conductor height and its interaction with infill were included to maintain the model hierarchy but were not significant by themselves.

Figure 13 shows the factorial plots associated with the gauge factor in the cTPU sensors. The infill angle had the largest impact, with the mean gauge factor in the 90° infill sensors reaching 27.12, which represented an increase of 114.6% over the 12.64 mean gauge factor found in the 0° infill parts. The layer height had a statistically significant but smaller contribution, showing a small increase in the gauge factor with increasing the layer height. The factorial plot for the conductor height showed very little impact, which was consistent with the effect being statistically insignificant.

## 4. Discussion

The initial resistance of both materials was strongly influenced by the height of the conductor. This was to be expected since the resistance of a theoretical ideal conductor
(2)$$R=\rho \frac{l}{A}$$
is linearly related to its cross-sectional area. In Equation (2), *R* is the resistance, ρ denotes the material resistivity, l defines the conductor length, and A is the conductor cross-sectional area [46]. However, the relationship between the resistance and the cross-sectional area in these AM parts was not linear, as would be the case for an ideal conductor. This was due to the fact that AM conductors are not ideal and homogeneous conductors. Inhomogeneities at the layer-to-layer and trace-to-trace interfaces in AM parts lead to a behavior that is better described as a network of conductors, resistors, and capacitors [47,48]. Because a conductor of a greater height has more layer-to-layer transitions, it exhibits a lower reduction in resistance than would be the case with a linear relationship. In the regression models, this behavior was captured by the effect of the conductor height squared. The interfacial resistance also caused *R*_0_ to be higher in the parts with a 0.1 mm layer height than in those with a 0.2 mm layer height, which the regression models captured as significant effects of the layer height and the interaction effect between the layer height and conductor height.

There were also differences in the initial resistance between the two materials. First, the cTPU parts had a higher resistance than the cPLA parts. This was due to the lower carbon black content in the cTPU material, see Table 2. This resulted in a less dense percolation network and therefore a higher resistance. Furthermore, the cTPU parts had a greater influence of the layer height and infill angle on *R*_0_ than the cPLA parts. In addition, the cTPU had the lowest resistance at a 0° infill angle, while the cPLA had the lowest resistance at a 45° infill angle. This, along with the more pronounced layer height effect, suggests that the cPLA material was better able to form connections across the intra- and interlayer interfaces. A 45° infill angle was then better at bypassing local imperfections in the cPLA. In contrast, the cTPU material exhibited higher resistance at the intra- and inter-layer interfaces, with the lowest resistance occurring at a 0° infill angle, where current flows primarily along continuous filament traces. If the effects of the layer height and infill angle alone are considered, selecting a favorable combination resulted in average initial resistances that were reduced by 22.3% and 66.5% compared to the combinations that led to the highest resistances for the cPLA and cTPU, respectively.

The gauge factor increased significantly for both materials as the infill angle was changed from 0° to 45° and from 45° to 90°. This suggests that the additional intra-layer interfaces created as the infill angle increased provided sites that were particularly sensitive to disruption in the percolation network. In addition, as the infill angle increased, these interfaces went from being parallel to the load direction at 0° to being perpendicular to the load at 90°. This increased sensitivity outweighed the disadvantageous effect of the infill angle on the initial resistance, as the 90° infill angle did not result in the lowest *R*_0_ in either material. It can therefore be argued that design approaches aimed solely at minimizing the unloaded sensor resistance [32,49] would miss significant effects. The behavior was stronger in the cTPU, which may be related to its lower ability to form bonds across the interfaces compared to the cPLA.

In the cTPU, there was a significant effect of the layer height and its interaction with other factors that was not present in the cPLA. The mean gauge factor increased with increasing the layer height. A potential explanation for this is that the effect of the layer height on *R*_0_ was stronger in the cTPU than in the cPLA. The additional layer-to-layer interfaces that were present at a reduced layer height led to an increase in *R*_0_, but this occurred on an axis that was not loaded during the tensile experiments. Therefore, these inter-layer interfaces increased *R*_0_ but did not act as sites that were particularly affected when the sensor was subjected to tensile loading, effectively leading to a decrease in the gauge factor.

There are a limited number of studies in the existing literature with which to compare these results. The work of Dijkshoorn et al. [14] reported a GF of 17.8 for the same cPLA material as used in this study, but the work was not specific about the infill angle of the sample measured. Therefore, the value can be said to be within the range found in the current work, but a direct comparison is not possible. There are also a number of works that have focused on the composition of conductive polymer [11,13,24], making the exact gauge factors reported difficult to compare with other materials. One such study, investigating a custom ABS material with a CNT filler [27], compared parts with a concentric infill pattern and a ±45° infill angle. The reported increase in GF from 4.94 to 7.99 was broadly consistent with the findings of the current work, although the absolute gauge factors were lower than those found in this study.

These findings bring the potential practical applications closer by identifying relevant parameters that can be used to achieve the desired sensor characteristics. Table 5 provides an overview of the typical design goals for each sensor characteristic and how the variables investigated in this work should be set to achieve that goal. Specifically, the lowest resistance for a sensor was achieved by maximizing the conductor height and layer height and setting the infill angle to either 0° for the cTPU or 45° for the cPLA. The highest gauge factor was achieved by increasing the infill angle and, in the case of the cTPU, increasing the layer height.

## 5. Conclusions

This work presented an investigation of the effects of the material, layer height, infill angle, and conductor height on the resistance and gauge factor in piezoresistive polymer sensors created by material extrusion AM. The results showed that for all the materials, the resistance was firstly influenced by the conductor height, with significant secondary influences from the layer height and the infill angle. The differences between the two materials appeared to be related to the conductive filler content and the ability to form low-resistance connections across the interfaces that exist both between and within the layers in AM parts.

The gauge factor was also significantly affected by the material and selected process parameters, while the sensor geometry had little effect. For both materials, the most significant increase in the gauge factor was achieved by increasing the infill angle. For the cTPU, there was also a statistically significant increase associated with increasing the layer height. The results of this work bring the potential practical applications closer, but are also of interest for future research in the field of additively manufactured piezoresistive sensors. Firstly, the precise documentation of the process parameters that have a significant effect on the sensor characteristics is necessary to describe experiments in such a way that they are reproducible and their results are comparable, which is not always the case in the current literature. Future research opportunities to extend this work lie in the identification of additional design and process parameters and their interactions that may influence sensor characteristics. Examples include the extrusion temperature, infill density, build orientation, or the use of more complex geometries such as meandering patterns. There are also opportunities to analyze other sensor characteristics, such as signal linearity, hysteresis, or repeatability under cyclic loading, and how these may be affected by process parameter selection. There are further opportunities for research into the integration of sensors into AM parts, such as the identification of optimal sensor locations or the establishment of external electrical connections to sensors embedded in a part. Finally, there are opportunities for the development of novel polymer composite materials optimized for sensing applications and compatible with MEX. While conductive filaments are commercially available, they are typically not specifically optimized for sensing and are often based on matrix polymers with a low heat deflection temperature, making them unsuitable for many applications.

## Figures and Tables

**Figure 1 polymers-15-02159-f001:**
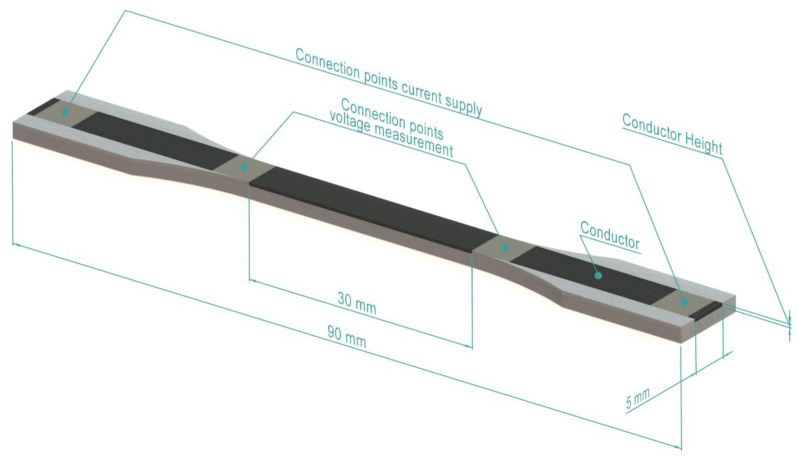
Rendering of a sensor, with the non-conductive TPU in white and the conductive element in black. The height of the conductive element varied for different variants. Dimensions not listed were in accordance with ISO 20753, variant A22 [41]. The connection points indicate the locations from which a four-terminal resistance measurement was taken.

**Figure 2 polymers-15-02159-f002:**
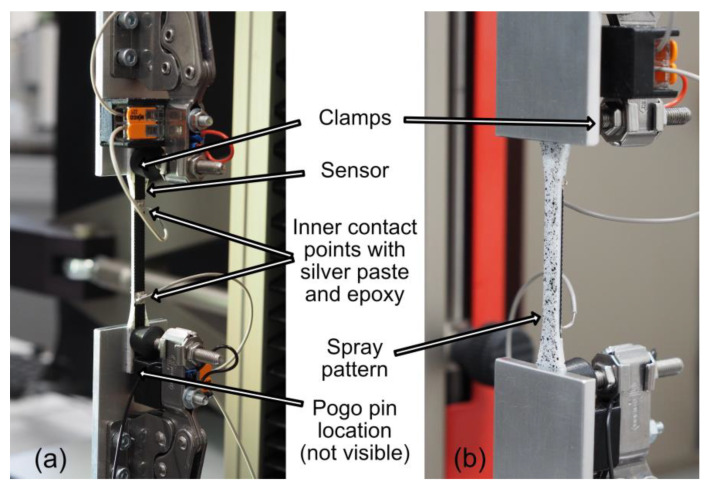
Photos of the measurement setup: (**a**) View from the conductor side, showing the inner two contact points for the four-terminal measurement. The outer two points are not visible because they were located under the clamping mechanism and were contacted by pogo pins. (**b**) Rear view of the sensor, showing the spray pattern used for video strain measurement.

**Figure 3 polymers-15-02159-f003:**
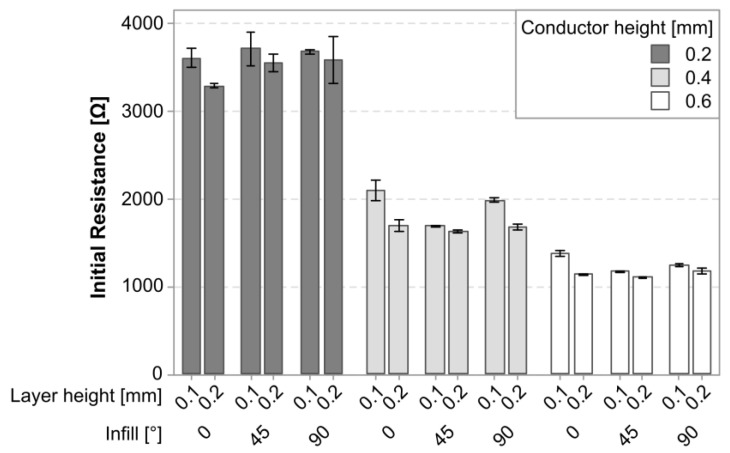
Plot of the mean initial resistance (*R*_0_) and its standard deviation in cPLA parts.

**Figure 4 polymers-15-02159-f004:**
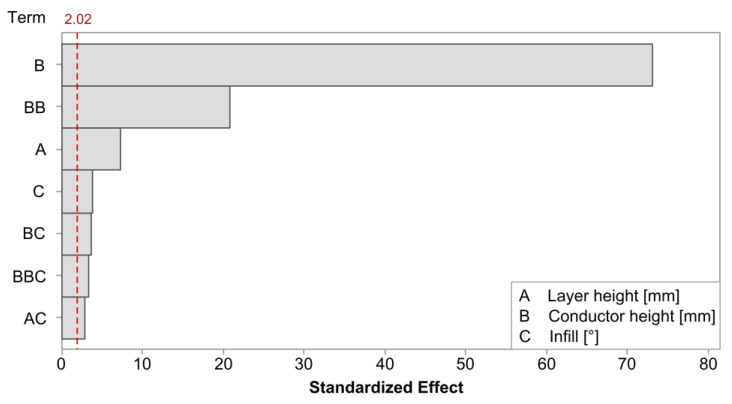
Pareto plot of the statistically significant terms on initial resistance in cPLA. The red dashed line represents a significance level of 0.05.

**Figure 5 polymers-15-02159-f005:**
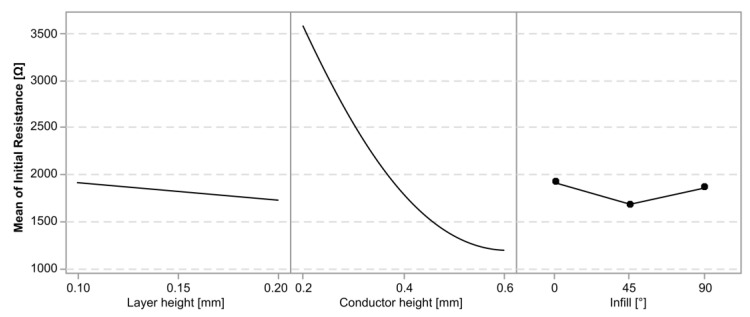
Factorial plots for the main effects on the initial resistance in cPLA sensors.

**Figure 6 polymers-15-02159-f006:**
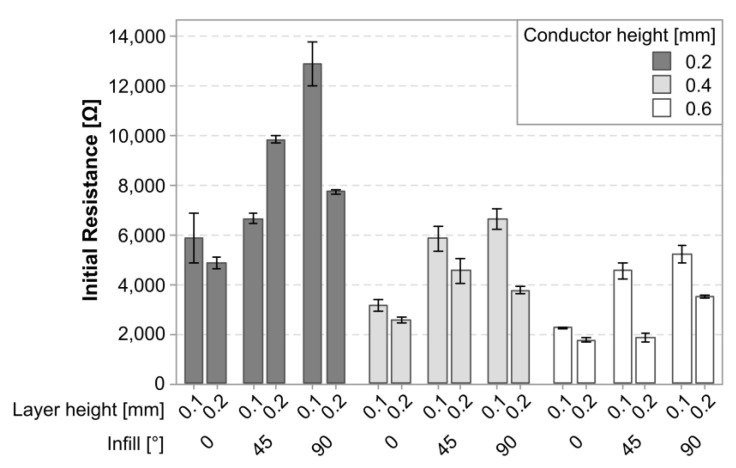
Graph of the average initial resistance (*R*_0_) and its standard deviation in cTPU parts.

**Figure 7 polymers-15-02159-f007:**
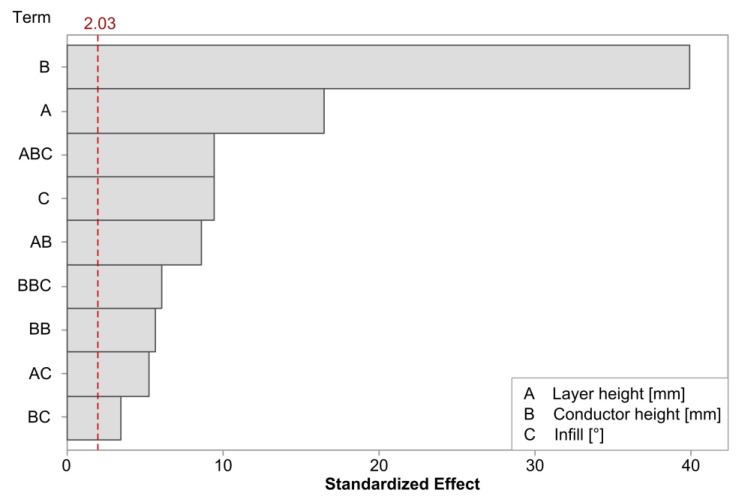
Pareto plot of the statistically significant terms on initial resistance in cTPU. The red dashed line represents a significance level of 0.05.

**Figure 8 polymers-15-02159-f008:**
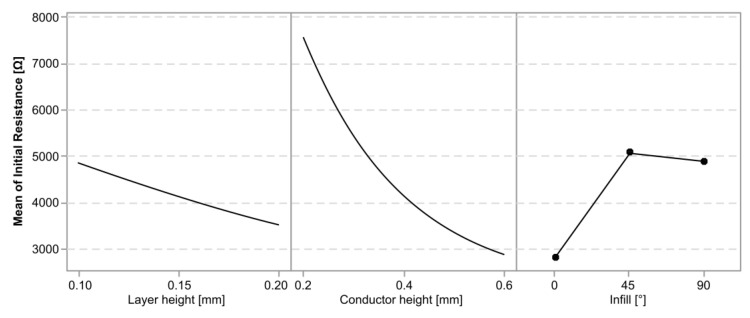
Factorial plots for the main effects on the initial resistance in cTPU sensors.

**Figure 9 polymers-15-02159-f009:**
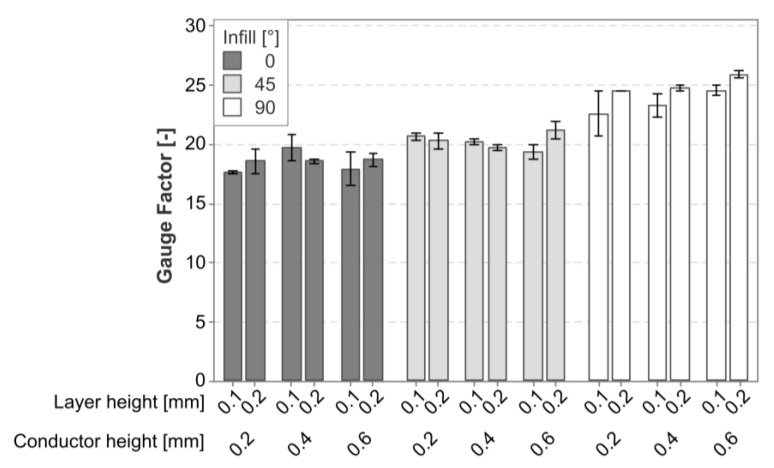
Graph of the mean gauge factor and its standard deviation in cPLA sensors.

**Figure 10 polymers-15-02159-f010:**
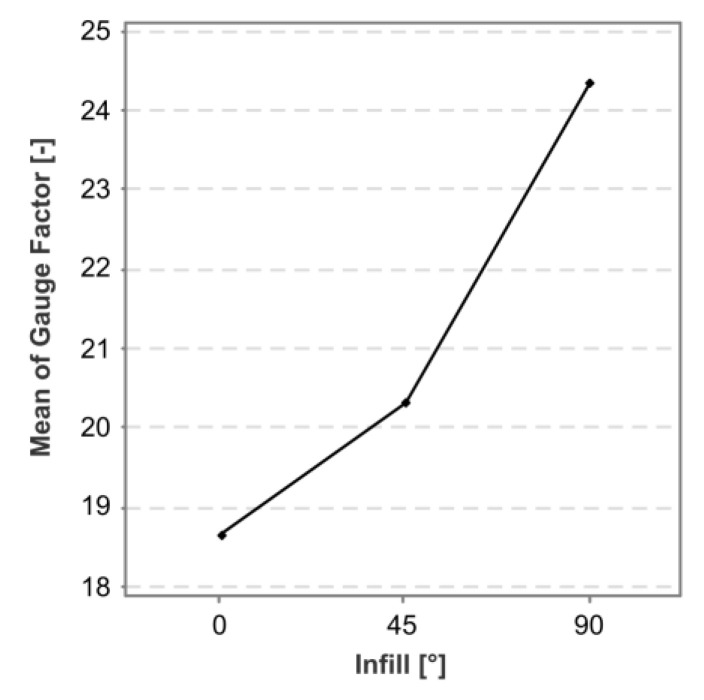
Factorial plot for the only significant effect for the gauge factor in cPLA sensors.

**Figure 11 polymers-15-02159-f011:**
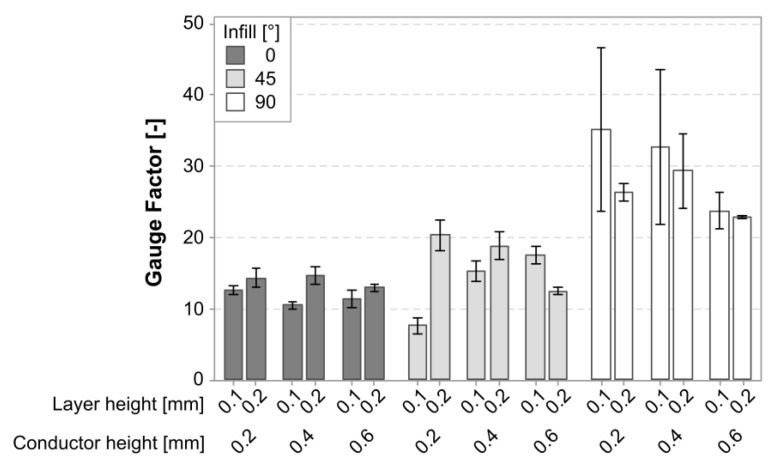
Graph of the mean gauge factor and its standard deviation in cTPU sensors.

**Figure 12 polymers-15-02159-f012:**
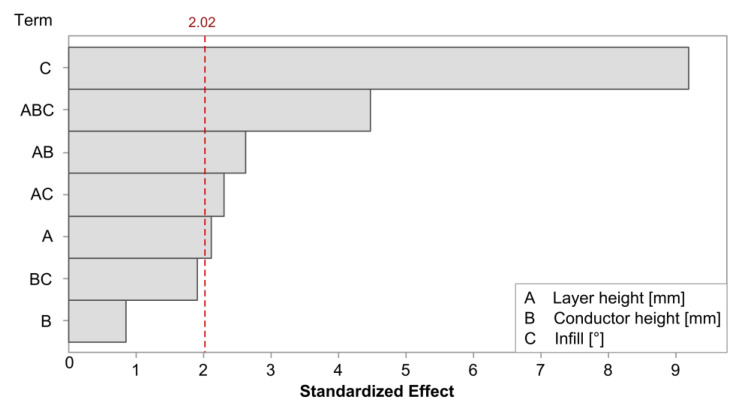
Pareto plot of the model terms for the gauge factor in cTPU. The red dashed line represents a significance level of 0.05. The model included two terms, labeled B and BC, which were not themselves statistically significant but were included to retain model hierarchy.

**Figure 13 polymers-15-02159-f013:**
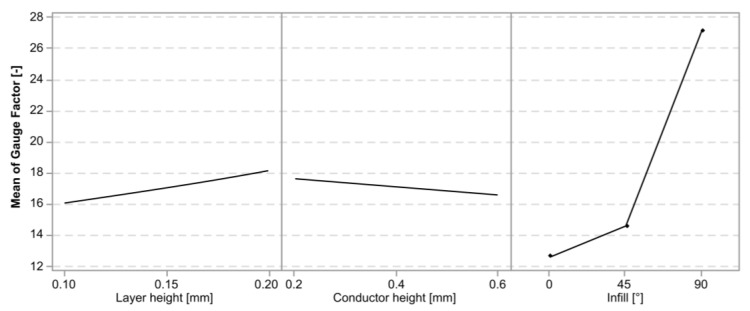
Factorial plots for the main effects on the gauge factor in cTPU sensors. The conductor height effect was not statistically significant but was included to retain model hierarchy.

**Table 1 polymers-15-02159-t001:** Overview of the existing literature on mechanical load sensors produced with MEX, separated according to the load cases and their research focus.

Load Case	Material Influences	Process Parameter Influences	Design/Geometry Influences
Tensile load	[11,13,24,25]	[14,26,27,28,29]	[12,28]
Compressive load	[7]	[26]	[22]
Flexural load	[7,25,30,31]	[31,32,33]	[30,34,35]

**Table 2 polymers-15-02159-t002:** Detailed information on the conductive filaments used.

Abbreviation	Polymer Matrix	Manufacturer	Filament Name	Conductive Filler	Resistivity
cPLA	Polylactic acid (PLA)	Proto-Pasta	Conductive PLA Filament [38]	Carbon black, ≤21.5% wt. [39]	*x*/*y* axis: 30 Ω·cm *z* axis: 115 Ω·cm
cTPU	Thermoplastic polyurethane (TPU)	NinjaTek	Eel 3D Printing Filament [37]	Carbon black, ≤18% wt. [40]	1.5 × 10^3^ Ω ^1^
TPU	Thermoplastic polyurethane (TPU)	NinjaTek	NinjaFlex [36]	None	Non-conductive

^1^ Volume resistance in accordance with ANSI/ESD STM 11.12, as provided by the manufacturer [37].

**Table 3 polymers-15-02159-t003:** Input factors used in the sensing element design and the additive manufacturing of the sensors. The layer height refers to the MEX process parameter, while the conductor height refers to the geometric design of the conductive part of the sensor, see Figure 1.

Variable Name	Experimental Settings
Conductive material	cPLA, cTPU
Conductor height	0.2, 0.4, 0.6 mm
Layer height	0.1, 0.2 mm
Infill angle	0°, ±45°, 90°

**Table 4 polymers-15-02159-t004:** Process settings used for each of the materials used in the experiment.

Setting	cPLA	cTPU	TPU
Temperature (°C)	235	235	235
Speed (mm/s)	15	20	20
Extrusion multiplier (-)	0.94	1.05	1.02
Trace width (mm)	0.4	0.4	0.4
Shells (-)	0	0	2

**Table 5 polymers-15-02159-t005:** Summary of the findings results, showing and how the experimental variables can be used to achieve the typical design goal for each sensor property. The most important variable influencing each property is shown in bold.

Sensor Property	Goal	Material	Sensor Height	Layer Height	Infill Angle [°]
Initial resistance	Minimize	cPLA	**Maximize**	Maximize	45 > 0 > 90
		cTPU	**Maximize**	Maximize	0 > 45 > 90
Gauge factor	Maximize	cPLA	Insignificant	Insignificant	**90 > 45 > 0**
		cTPU	Insignificant	Maximize	**90 > 45 > 0**

## Data Availability

The data presented in this study are available in Appendix A.

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
