# Peer review of "Process Parameters and Geometry Effects on Piezoresistivity in Additively Manufactured Polymer Sensors"

_polymers, 2023, doi:10.3390/polym15092159_

Round 1
Reviewer 1 Report
Review of the paper entitled
“Process parameter and geometry effects on piezoresistivity in additive manufacturing polymer sensors”
The paper is well-written and clear. The developed knowledge is useful for the scientific community. The literature review and state-of-the-art are well presented. The references are in adequate number and type. The structure of the paper is well-organized such that the information flows in a logical manner. The purpose of the study is clearly outlined. The results and discussion support the conclusions.
Some minor comments:
· Title: Consider plural for “process parameter”.
· Some quantitative results should be anticipated in the abstract.
· Rows 54-55. Some additional comments about the compatibility of polymer matrixes and fillers would be appreciated. Which requirements make a filament matrix feasible for this application? i.e., matrix-filler adhesion, filler type (fiber, particles, etc.), processability, etc.
· “Introduction” section. The state-of-the-art is complete, but a recap of some quantitative results would be appreciated. For example, a table summarizing the effects of MEX technology process parameters on sensors performance and fabrication.
· Table 2 should be improved in format.
· Table 2. If possible, please report the electrical conductivities of the materials.
· Table 2. Reference [32] reports a link to the safety data sheet of the material. Reporting the web link to the technical (not safety) datasheet is suggested, as correctly done in reference [31].
· Table 4. From the supplier datasheet, the suggested extrusion temperatures are 215°C and 220-230°C for cPLA and cTPU, respectively. The extrusion temperature set to 235°C for both materials can enable different cross-linking connections in the different materials. This value seems to be the upper limit for both materials. Some comments about the choice of this value of temperature are required. As correctly stated by the authors in the conclusions (rows 431-432), the effects of the extrusion temperature should be investigated.
· Table 4, the authors should clarify their choice in the invariant process parameters.
· Row 212. The equipment is not accurately identified. It is required to report the model and its accuracy.
· Red dashed lines in all graphs are not visible o too clear or behind bars.
· The “Supplementary materials” section has not been filled.
· Typo: “al” at row 174.
Reviewer 2 Report
Overall, this is a good paper. Two revisions are required.
3.2.3. Discussion Gauge Factor section is just on Gauge Factor. That section should just be on Discussion. Gauge Factor should be deleted.
4. Conclusions section needs major changes 1) Major cuts are required. Almost 70% percent of the current conclusion should eb deleted. 2) Remove Table 5 from the Conclusions. 3) You can not have any reference, table and figure in Conclusions.
Reviewer 3 Report
It is an original paper dealing with “Process parameter and geometry effects on piezoresistivity in additive manufactured polymer sensors “.Regarding this manuscript there are some minor and major comments below to help the readers to be more beneficial from the paper.
1. The abstract is written as a general description. It should describe your main achievements, and results.
2. In line 36, the author need to more referred to the MEX process as it can be used for recycling of composite materials.
[a] "Manufacture of Composite Filament for 3D Printing from Short Glass Fibres and Recycled High-Density Polypropylene," in Proceedings of the World Congress on Mechanical, Chemical, and Material Engineering, 2022.
[b] "Recovery of Particle Reinforced Composite 3D Printing Filament from Recycled Industrial Polypropylene and Glass Fibre Waste," Proceedings of the World Congress on Mechanical, Chemical, and Material Engineering, 2022.
3. In Line 58, the author says “Conductive polymer composites can be used to fabricate a wide variety of electronics and sensors “.Refer to the following references for further information related to peizoresitive sensors
[c] Estimating ground reaction force with novel carbon nanotube-based textile insole pressure sensors. Wearable Technologies, 4, e8.
[d] Piezocapacitive sensing for structural health monitoring in adhesive joints." In 2019 IEEE International Instrumentation and Measurement Technology Conference (I2MTC), pp. 1-5. IEEE, 2019.
4. Line 174, in “all samples” is true
5. In section 2.2, the load rate during the tensile test is not indicatde
6. The authour need to use a nonconductive endtabs to record the electrical resistance changes between silver connection points, otherwise the sensor output doesn’t show the real electrical changes during the tensile test
7. Table 5 should be moved to the previous section
8. The conclusion is the summary of your main results. If you need to refer to references, they must be included in the discussion section. Use bullets in Conclusions to emphasise the main achievements of the paper
Round 2
Reviewer 3 Report
In comment 2, the authors refer AM and MEX in general and do not mention piezoresistive materials specifically. In addition, the added reference [4] does not show MEX filament preparation and recycling procedure for 3D printing since the material is commercially available already in this reference. The authors should include two references [a] and [b] as already mentioned.
[a] "Manufacture of Composite Filament for 3D Printing from Short Glass Fibres and Recycled High-Density Polypropylene," in Proceedings of the World Congress on Mechanical, Chemical, and Material Engineering, 2022. [b] "Recovery of Particle Reinforced Composite 3D Printing Filament from Recycled Industrial Polypropylene and Glass Fibre Waste," Proceedings of the World Congress on Mechanical, Chemical, and Material Engineering, 2022.
Round 3
Reviewer 3 Report
Accept in present form